# Hydrological–Thermal Coupling Simulation of Silty Clay during Unidirectional Freezing Based on the Discrete Element Method

**Wei Shan** [1,2,3,4,*] , **Shiyao Qu** [1] **and Ying Guo** [1,2,3,4]

1   Institute of Cold Regions Science and Engineering, Northeast Forestry University, Harbin 150040, China;
    ying-guo@nefu.edu.cn (Y.G.)
2   Ministry of Education Observation and Research Station of Permafrost Geo-Environment System in Northeast
    China (MEORS-PGSNEC), Harbin 150040, China
3   Collaborative Innovation Centre for Permafrost Environment and Road Construction and Maintenance in
    Northeast China (CIC-PERCM), Harbin 150040, China
4   Low-Carbon Road Construction and Maintenance Engineering Technology Research Center in Northeast
    Permafrost Region of Heilongjiang Province (LCRCMT-HLJ), Harbin 150040, China
*   Correspondence: shanwei456@163.com

**Abstract:** A hydrological–thermal coupling discrete element model depicting the unidirectional freezing process of unsaturated silty clay was developed in order to investigate the migration law of unfrozen water in unsaturated silty clay under unidirectional freezing circumstances. The model uses the contact heat transfer equation to calculate the heat transfer process while taking into account the latent heat of phase transition. To obtain the silty clay's freezing characteristic curve, the model combines the unfrozen water content curve with the Clausius–Clapeyron equation. The water migration from the unfrozen zone to the frozen zone was calculated using Harlan's model and the frozen fringe hypothesis. The discrete element application MatDEM 3.0 was used to incorporate the mathematical model for computation, and the output was compared to the result of indoor unidirectional freezing tests. The soil closest to the stable freezing front had the largest water content, according to the findings of numerical modeling and laboratory testing, and unfrozen water in the soil would move from the unfrozen zone to the frozen zone under the action of water potential difference. The results of laboratory tests and numerical simulations can accurately describe the temperature variation and water migration of soil during freezing, demonstrating the accuracy of the established discrete element model and proving the viability of the discrete element method in the study of frozen soil.

**Keywords:** discrete element method; hydrological–thermal coupling; numerical analysis; unidirectional freezing

## 1. Introduction

Permafrost is ground that remains at or below 0 °C for at least two consecutive years, and seasonally frozen ground is ground that freezes and thaws annually [1]. In China, the permafrost region, which covers an area of 1.59 million square kilometers and accounts for 16.5% of the country's total land area, is found in the Qinghai–Tibet Plateau, Qilian Mountains, Tianshan Mountains, Altai Mountains, Greater and Lesser Khingan range, and other high mountains or high-latitude areas [2]. However, the majority of Inner Mongolia and China's northeast, north, and northwest regions are home to seasonally frozen ground regions. The average yearly temperature, height, and other factors affect the active layer's depth, which can be up to 3 m [3]. Frozen soil is a complex, non-homogeneous, anisotropic, multiphase material consisting of four primary components: solid soil particles, ice, liquid water, and air. The water in frozen soil constantly experiences phase change, whether it is in a permafrost or seasonally frozen ground region. This causes the corresponding frost

damage, such as frost heave, thaw settlement, and flood-plain icing. These risks are all connected to the thermo–hydro–mechanical (THM) coupling process and the phase change of water. It is essential to study the multi-field coupling function during the phase change of water.

Harlan [4] initially proposed a hydrological–thermal coupling theoretical model to study the freezing process of soil in multi-field coupling investigations of frozen soil, and he considered the water potential as the driving force for water migration as estimated with the Clausius–Clapeyron equation. Harlan's model was enhanced by Newman and Wilson [5], who determined water and ice contents of frozen soils using a permeability coefficient and suction function without an impedance coefficient. By examining the heat transport terms in Harlan's model, Nixon [6], Taylor, and Luthin [7] were able to simplify the heat equation.

O'Beil and Miller [8] created the rigid ice model, proposed the concept of frozen fringe, and highlighted how the freezing front is distributed in frozen soil and how it relates to the ice lens. Konard and Morgenstern [9] noted that the water migration rate of the warm end of the end ice lens is proportional to the temperature gradient in the unfrozen zone when the end ice lens enters a thermally stable condition, and this ratio coefficient is segregation potential. Based on the adsorption membrane theory, Chen et al. [10] developed a water driving force model for frozen soil that connected temperature, unfrozen water content, and pressure between ice and water phases in pore space. According to Dash and Rempel [11], during freezing, unfrozen water present around soil particles migrates into the surrounding ice crystals. Ming et al. [12] introduced the concept of migration potential and developed a semi-empirical water migration model.

When conducting numerical simulations of multi-field coupling during soil freezing, based on the theory of seepage and heat conduction in unsaturated soil, Bai et al. [13] developed a joint solution equation for the hydrological–thermal coupling problem in frozen soil. Subsequently, they employed COMSOL Multi-physics software to conduct a fully coupled numerical simulation of temperature and moisture fields in frozen samples. Li et al. [14] carried out numerical simulations of frost heave and studied the connection between unfrozen water migration and soil's temperature. Based on physical and geotechnical testing and field monitoring data, Hu et al. [15] and Guo et al. [16] conducted statistical modeling of the vertical displacement of two roadbed sections, namely K161+440 and K181+400, along the Beian–Heihe highway.

The finite element method (FEM) used in the abovementioned studies is a well-established and widely applicable numerical method that is often used to investigate soil multi-field coupling. It has also been continuously developed and improved by numerous academics. Rocks and soils, on the other hand, are porous systems at the micro level that are relatively continuous and made up of particles, solutes, and cracks. Methods based on the continuous medium have limitations when dealing with problems that involve discrete characteristics and discontinuities in rocks and soils. For instance, the abovementioned methods cannot account for the effect of pores between soil particles on heat transfer. The continuous methods have difficulty producing accurate calculations in the case of microstructure changes brought on by variations in the physical characteristics of soil particles before and after phase change of water in soil. For example, models capable of multi-field coupling calculations have been developed [13,17], which have been simplified to some extent, for the unfrozen water migration during phase change. Therefore, discontinuous medium methods will be more practical for future studies of more intricate issues. There are various application scenarios for different models. For example, the rigid ice model is used to estimate the expansion of the ice lens during frost heave, and Harlan's model is appropriate for the computation of hydrological–thermal coupling and frost heave in engineering. These ground-breaking methods have advanced considerably until now. The process of creating numerical models has also advanced at the same time.

At present, some scholars have used the discrete element method (DEM) to model the multi-physical field process during soil freezing. Using PFC software, An et al. [18]

studied the macroscopic mechanical properties and the damage mechanism of frozen soil, as well as explored the FDM-DEM model to investigate the fine-scale mechanism of roadbed deformation in the seasonal frozen soil zone. Yin et al. [19] established a discrete-element numerical model of frozen clay based on the three-dimensional particle DEM and performed triaxial simulation tests. Ding [20] et al. studied the relationship between the strength, elastic modulus, and curing period of cement-modified frozen soil based on DEM. The abovementioned studies explained the mechanical properties and damage mechanisms of frozen soil at the microscopic level from different perspectives and introduced the modeling methods of DEM to the studies of frozen soils. Although there are limited studies on the use of DEM for modeling unfrozen water migration in frozen soil, the studies achieved by Zhang et al. [21], Sang et al. [22], and Trans et al. [23] using DEM for modeling provide ideas for the work in this paper.

In this study, the contact heat transfer equation and the water migration model were used to simulate the indoor unidirectional freezing tests of unsaturated silty clay with the discrete element program MatDEM 3.0. This study described the variations of soil temperature and water content through indoor tests and numerical simulations, verifying the feasibility of the discrete element hydrological–thermal coupling model and providing ideas for the application of DEM in frozen soil studies.

## 2. Modeling Ideas and Assumptions

The hydrological–thermal coupling model of frozen soils is a highly relevant and current topic of research. There exist various theoretical frameworks and theories in this field. Based on the frozen fringe theory and Harlan's model, this article developed a discrete element hydrological–thermal coupling model for use during the freezing process of unsaturated silty clay. Miller [24] emphasized that ice particles develop in the pores beneath the warmest ice lens. This region, which has low permeability, low water content, and no frost heave, is known as "frozen fringe". According to the frozen fringe theory, the soil columns in unidirectional freezing tests were divided into three zones: frozen zone, frozen fringe, and unfrozen zone. The temperature of the warm end of the active ice lens is the segregation temperature $T_s$, the temperature at the freezing front is the freezing temperature $T_f$, and the frozen fringe is located within this temperature range, as illustrated in Figure 1.

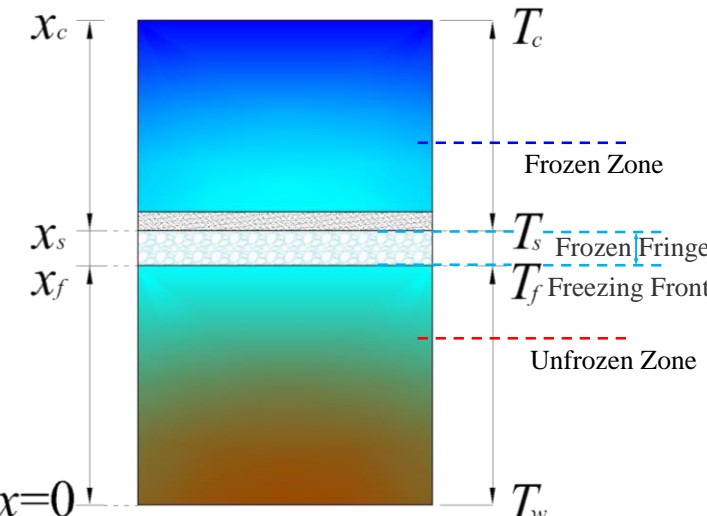

**Figure 1.** Schematic diagram of frozen fringe theory.

In the unidirectional freezing test, the freezing front is constantly moving, and the formation and development of the ice lens are also dynamically changing. Konrad and Morgenstern [25] used thermal physical methods to judge that the segregated ice was produced at the temperature $T_{sf}$ in the frozen fringe. The condition for stopping growth is

that the secluding temperature is lower than $T_{sm}$. Zhou et al. [26] observed the secluding temperature of the end of the ice lens in a continuous freezing test. The results showed that the segregation temperature of the end ice lens decreased slightly with the increase of freezing time.

The experiment described in ref. [27] shows that the water migration in the area above the active ice lens is low, and the water migration during soil freezing process mainly occurs in the frozen fringe. The intense water migration process also promotes the growth and frost heave of ice lens. The bottom end of the ice lens will generate a negative equivalent pressure. Under the action of this negative equivalent pressure, the water in the unfrozen zone will migrate to the bottom end of the ice lens. The driving force of water migration is the water potential in Harlan's model.

Frozen soil, being a porous medium, provides multiple pathways for water migration. These include the following: (I) water vapor diffusion through the soil pore system; (II) liquid water migration through capillaries within the pores and water vapor condensation at the capillary meniscus; and (III) the flow of liquid water along a continuous water film adhered to soil particles [28]. Based on frozen fringe theory and different modes of water migration, the following assumptions were used to establish the model.

Assumption 1: A series of particles is used to represent a soil with a certain water content (Figure 2a). Each particle element has its own property parameters (volume water content, volume ice content, etc.) and occupies a certain equivalent volume. In this paper, the particles in the soil were assumed to be spheres, and the DEM model shown in Figure 2b was established. The volume occupied by the particle elements and the water around them is represented by the equivalent volume $V_i$ (Figure 2c).

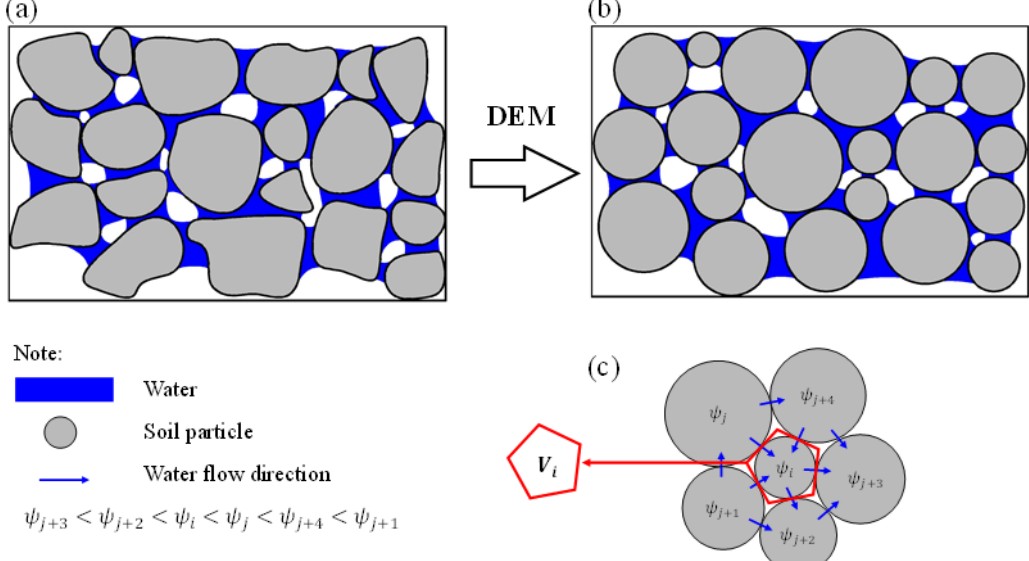

**Figure 2.** (**a**) The arrangement of particles and water distribution in soil; (**b**) equivalent DEM model; and (**c**) a schematic diagram illustrating water migration between particles.

Assumption 2: Unsaturated seepage will occur between adjacent particles if their water potentials differ from one another. The water migration process follows Darcy's law, and water flow will migrate from the particle elements with high water potential to those with low water potential (Figure 2c). Similar to this, contact heat transfer will take place between neighboring particles when their temperatures are different, causing heat to transfer from high-temperature components to low-temperature components.

Assumption 3: Freezing temperature $T_f$ at the freezing front determines the frozen fringe's lower border, whereas the warmest ice lens segregation temperature $T_s$ determines its upper boundary. Frost heave brought on by the rise of the ice content of soil particles

is not taken into account, and neither are water migrations in the frozen zone nor water vapor movement inside samples.

## 3. Theories and Methods

### 3.1. Particle Contact Model

The discrete element method (DEM) was initially introduced by Cundall and Strack [29] as a tool to investigate the motion and interactions of granular materials. DEM models the behavior of individual particles, which are assumed to be independent of one another, and interact through contact, resulting in the generation of forces and displacement. The DEM can also account for the water migration and heat transfer in soil, as these processes are linked to the contact of the particle elements. In this study, MatDEM 3.0 software developed by Nanjing University was employed.

In MatDEM 3.0, a linear elastic model was used for the contact mode of the particle elements. The software describes the cementation that exists between soil particles by introducing normal and tangential springs in the simulation. Cementation represented by spring equivalents is expressed as normal stiffnesses $K_n$ and tangential stiffnesses $K_s$, which are automatically calculated in the software based on the contact state between particle elements.

In this paper, since radius change and the mechanical influence of the element after frost heave have not been deeply studied, the contact model was only used to calculate the contact force between elements.

### 3.2. Hydrological–Thermal Coupling Model

The changes in water content and temperature in soil will affect each other. Harlan [4] put forward the "hydrodynamic model" based on the latent heat of phase transition and the law of mass conservation and hydrodynamics.

$$\frac{\partial}{\partial x}\left(\lambda_f \frac{\partial T}{\partial x}\right) - C_w \rho_w \frac{\partial(V_w T)}{\partial x} = \overline{C_\rho}\frac{\partial T}{\partial t} - L_f \rho_i \frac{\partial \theta_i}{\partial t} \tag{1}$$

$$\rho_w \frac{\partial}{\partial x}\left(K_f \frac{\partial \varphi}{\partial x}\right) = \rho_w \frac{\partial \theta}{\partial t} + \rho_i \frac{\partial \theta_i}{\partial t} \tag{2}$$

In the above-shown equation, $C_w$ is the volume heat capacity of water, $\overline{C_\rho}$ is the volume heat capacity of soil, $L_f$ is the latent heat of phase change of water, $K_f$ is the permeability coefficient, $\rho_i$ and $\rho_w$ are the densities of ice and water, respectively, $T$ is the Kelvin temperature of soil, $\varphi$ is the total pressure of the water head, $\theta$ and $\theta_i$ are volume water content and volume ice content, respectively, and $V_w$ is the flow rate of the water.

Taylor and Luthin [7] pointed out that the heat change caused by water migration only accounts for 1‰~1% of the total heat, so the second item in Equation (1) can be ignored. Equation (2) is used to calculate the amount of water that migrates from the unfrozen area to the frozen area under the action of the negative equivalent pressure formed by the ice lens.

The water potential used to calculate the total pressure head $\varphi$ in Harlan's model can be solved using the Clausius–Clapeyron equation. Groenvelt and Kay [30] described the relationship between temperature $T$ and water potential of frozen soil according to the Clapeyron equation as follows:

$$\psi = L_f ln\left(\frac{T}{273.15}\right) \tag{3}$$

Since the freezing temperature of the water in soil is usually below 0 °C, the water potential of frozen soil is expressed using a modified version of the above-shown equation:

$$\psi = L_f ln\left(\frac{T - T_0}{273.15 - T_0}\right) \tag{4}$$

In the above-shown equation, $T_0$ is the Kelvin temperature of the soil at freezing temperature, and the total pressure head $\varphi$ is calculated in Section 3.2.2.

Since the phase change of water during soil freezing is not a transient process, the water content at a certain temperature is not a constant value, but a function of the temperature gradient and time. In order to determine the water potential of the soil at different locations during soil freezing, it is necessary to characterize the soil's water holding properties during the freezing process.

The freezing characteristic curve of soil can be determined by employing the method of Croney et al. [31], which utilizes the unfrozen water content curve and the Clapeyron equation to characterize the water holding properties of frozen soil. The Anderson and Tice [32] model was used to fit the relationship between unfrozen water content and temperature:

$$\theta_u = \alpha |T - T_0|^\beta \tag{5}$$

In the above-shown equation, $\theta_u$ is the volume content of unfrozen water and $\alpha$, $\beta$ are fitting parameters.

Following the aforementioned processing, a series of data on $\theta_u$ and $\psi$ are obtained. Finally, the relationship between water potential and unfrozen water content during soil freezing can be determined by fitting the data to the Fredlund-Xing's three-parameter model [33], as shown in Equation (6).

$$\frac{\theta}{\theta_s} = \frac{1}{\left( ln \left( e + (\psi/a)^n \right) \right)^m} \tag{6}$$

In the above-shown equation, $\theta_s$ is the saturated volume water content and $a$, $m$, and $n$ are fitting parameters.

Harlan's model and its enhanced model are frequently employed in the continuous medium method. However, only a few similar models have been researched and developed using the DEM to simulate the hydrological–thermal coupling process. The following section discusses the heat transfer and water migration processes in the discrete-element hydrological–thermal coupling model.

3.2.1. Heat Transfer Model

According to the Fourier equation, Zhang et al. [34] applied the theory of heat transfer in a continuous medium to the DEM.

$$Q = \frac{kA\Delta T}{L} \tag{7}$$

In the above-shown equation, $Q$ is heat flow, $k$ is thermal conductivity, $A$ is cross-sectional area, $\Delta T$ is the temperature difference between heat sources, and $L$ is the spacing of heat sources.

In a macroscopic scale, the Fourier equation is appropriate for calculating heat transfer in a continuous medium with a regular shape, However, for soil with pores inside, this method is not suitable for more refined calculations. Therefore, Vargas and McCarthy [35] incorporated the contact heat transfer equation [36] into the heat transfer calculations of the DEM (Equations (8) and (9)):

$$\rho_i C_i V_i \Delta T_i = \sum_{j=1}^{N} h_{ij} \left( T_j - T_i \right) \tag{8}$$

$$h_{ij} = k_s \left( \frac{3F_n r}{4E^*} \right)^{1/3} \tag{9}$$

In the above-shown equation, $i$ and $j$ are the numbers of adjacent particle elements, $\rho_i$ is the density of the soil, $C_i$ is the volume heat capacity of soil, $V_i$ is the equivalent volume

of a particle element, $\Delta T_i$ is the temperature change of a particle element, $h_{ij}$ is the thermal contact resistance between the particles, $F_n$ is the contact forces between the elements, $E^*$ is the effective elastic modulus, $r$ is the effective contact radius, and $k_s$ is the thermal conductivity.

The parameter $k_s$ in Equation (9) determines the value of the thermal conductivity. For the contact heat transfer of two adjacent elements, it is inappropriate to use measured thermal conductivity, because the thermal conductivity measured in the laboratory contains the comprehensive thermal conductivity of the air in the soil pores, and it is not the thermal conductivity of the contact between the soil particles. Therefore, the Maxwell–Eucken equation [37] is introduced here to calculate the real thermal conductivity of the material when the particle elements are in contact (Equation (10)).

$$K = \frac{k_1 v_1 + k_2 v_2 \frac{3k_1}{2k_1+k_2}}{v_1 + v_2 \frac{3k_1}{2k_1+k_2}} \tag{10}$$

In Equation (10), $K$ is the experimentally measured thermal conductivity, $k_1$ is the thermal conductivity of the components with more volume content in the two-phase mixture, and $k_2$ is the thermal conductivity of the components with less volume content in the two-phase mixture. $v_1$ and $v_2$ are the ratio of soil and air. $k_1$ is the actual thermal conductivity of the soil particles, $k_1 = k_s$.

Because the content of each component of soil will change constantly during the freezing process, ignoring the influence of air on the specific heat of the soil, the specific heat $C_i$ of the soil is expressed by Equation (11):

$$C_i = C_s n + C_w (1-n)(1-\theta_{ice}) + C_{ice}(1-n)\theta_{ice} \tag{11}$$

In Equation (11), $n$ is the proportion of soil particle volume occupying the soil, $C_s$ is the heat-specific volume of the soil skeleton, and $C_{ice}$ is the heat-specific volume of ice.

The latent heat will be released when the soil freezes, and the temperature change of the soil elements in a time step can be calculated using Equation (12):

$$\Delta T_i = \Delta t \left( \sum_{j=1}^{N} \left( \frac{h_{ij}(T_i - T_j)}{C_i m_i} \right) + L_f \frac{\Delta m_{ice}}{C_i m_i} \right) \tag{12}$$

In Equation (12), $C_i$, $m_i$ and $T_i$ are the specific heat, mass, and temperature of the soil element $i$, respectively, $T_j$ is the temperature of the soil element neighboring the soil element $i$, and $\Delta m_{ice}$ is the change rate of ice mass.

### 3.2.2. Water Migration Model

Soil is a porous medium, and DEM can treat a certain volume of porous media as an element. Trans et al. [23] applied the water migration theory in a continuous medium to simulate the water transmission process of sand with the water content difference as the driving force. However, because of the lower water potential of clay, the water content in clay cannot be calculated with the water content difference as the water driving force. Although Trans's model cannot be directly used in the study of unsaturated clays, it is an important idea to use the water content difference as the water driving force and use the difference idea to calculate the water migration in discrete element model. The equation used to calculate the change in the water content of unsaturated sand in unit time $\Delta t$ is as follows:

$$\Delta \theta_i = \Delta t \left( \frac{1}{V_i} \sum_{j=1}^{N} d_{ij} \frac{\Delta \theta_{ij}}{L_{ij}} \right) = \Delta t \left( \frac{1}{V_i} \sum_{j=1}^{N} Q_{ij} \right) \tag{13}$$

In Equation (12), $\theta_i$ is the volume water content of the soil particles $i$, $V_i$ is the efective volume occupied by the soil particles, $N$ is the number of adjacent particles to the soil

particles $i$, $d_{ij}$ is the microscopic diffusion coefficient, and the water content difference between the soil particles $i$ and its adjacent particles $j$, $L_{ij}$ is the distance between the particles $i$, $j$, and the terms in parentheses are the migration flux in unit time $\Delta t$.

In the frozen soil, the water migration in the unfrozen zone can still be regarded as a Darcy flow under the action of the pressure potential layer. Under the action of the pressure layer induced by the temperature layer, the water migration rate $v$ and the migration flux $Q$ in the unfrozen area can be expressed as follows:

$$v = \frac{k\varphi}{l} \tag{14}$$

$$Q = \frac{kA\varphi}{l} \tag{15}$$

In the above-shown equation, $k$ is the permeability coefficient, $l$ is the seepage length, and $\varphi$ is the total head pressure, which can be expressed as follows:

$$\varphi = \frac{\psi}{\rho_w g} \tag{16}$$

When the soil is frozen, the increase of pore ice will hinder the migration of unfrozen water, so the ice block coefficient $I$ is introduced. Combined with Equation (14) and frozen fringe theory (Figure 1), the water migration rate of the frozen zone, frozen fringe, and unfrozen zone can be expressed as follows:

$$v = \begin{cases} \frac{kI\varphi}{l}, & x < x_f \\ \\ 0, & x \geq x_s \end{cases} \tag{17}$$

$$I = 10^{F\theta_{ice}} \tag{18}$$

In Equation (18), the empirical constant usually takes the value $-10$.

The freezing and melting of water are non-transient processes. The macroscopic performance of crystal growth is the change in ice water volume in the granular medium. Koop et al. [38] and others have proposed a "water activity standard" for the homogeneous nucleation of ice. The water activity as described by the macroscopic ice water phase transition rate [39] and the change in ice content in the granular element volume can be expressed as follows:

$$\dot{m} = K(T^*)_f (1 - a_w)^{n_{wi}} c_w \tag{19}$$

$$\Delta\theta_{ice} = \frac{\dot{m} V_i}{m_i} \tag{20}$$

In the above-shown equation, $K(T^*)_f$ represents the forward constant of freezing, $a_w$ is water activity, $n_{wi}$ is the material parameter, and $c_w$ is the concentration of water.

The equation of the change of the volume water content of the particle (Equation (21)) in the discrete element model within a unit time can be obtained using Equations (2), (13), (17) and (20):

$$\Delta\theta_i = \Delta t \left( \frac{1}{V_i} \sum_{j=1}^{n} \left( \frac{kAI\varphi}{l} \right) - \frac{\rho_w}{\rho_i} \Delta\theta_{ice} \right) \tag{21}$$

## 4. Materials and Tests

In this chapter, the flow of indoor tests (Sections 4.1–4.3) and simulation tests (Sections 4.4 and 4.5) are mainly introduced.

*4.1. Indoor Tests Material*

The soil samples used in the tests were taken from Harbin, Heilongjiang Province, China.

### 4.1.1. Dry the Soil

Specified quantities of soil samples were placed onto trays and then subjected to an oven drying process at a temperature of 105 °C. After drying, the soil samples were crushed and screened in layers with a maximum aperture of 2 mm. The sieved soil samples were placed back in the oven for further drying to prevent water absorption during the resting process. The particle size distribution of the soil samples is shown in Table 1.

**Table 1.** Particle size distribution of the soil samples.

| Particle Size Range | 1~2 (mm) | 0.5~1 (mm) | 0.25~0.5 (mm) | 0.15~0.25 (mm) | 0.075~0.15 (mm) | <0.075 (mm) |
|---|---|---|---|---|---|---|
| Percentage | 2.5% | 12.6% | 9.7% | 16.0% | 21.1% | 34.1% |

### 4.1.2. Configure the Soil

The dried soil sample was removed from oven, and the required mass of dry soil and water was calculated based on the specific test design. A certain mass of dry soil sample was placed onto a tray, and distilled water was uniformly sprayed onto it. In the process of adding distilled water, careful stirring was required, and the prepared soil samples were placed in a sealed bag for 24 h to ensure that water was uniformly distributed throughout the soil samples.

### 4.1.3. Subsubsection

The soil sample was layered and compacted inside the sample cylinder. After pressing from the top, a cylindrical sample measuring 75 mm in diameter and a 150 mm in height was obtained.

*4.2. Tests Program*

The objective of the tests was to investigate the temperature and water content changes in a soil column under axial freezing in a closed system. The test was conducted on clay samples with an initial mass water content of 20%. Various temperature gradients were set up to observe the effects on the soil column. The test conditions are presented below (Table 2).

**Table 2.** Unidirectional freezing test conditions.

| Test Number | Initial Mass Water Content | Top Plate's Temperature (°C) | Bottom Plate's Temperature (°C) |
|---|---|---|---|
| 1 | 20% | −5 | 1 |
| 2 | 20% | −7 | 1 |
| 3 | 20% | −10 | 1 |

*4.3. Tests Method and Procedures*

The test equipment consisted of four main components: cold liquid circulators, water bath plates, temperature sensors, and a data acquisition system. The overall structure of the test equipment is shown in Figure 3 To record the temperature change of the sample when it was frozen in a unidirectional direction, thermistor-type temperature sensors were placed at specific heights (0.14 m, 0.12 m, 0.10 m, 0.08 m, 0.06 m, and 0.04 m) within the sample. The temperature sensor used in the indoor test is the WZP-GZPT-A thermistor produced by Hangzhou Guizhong Technology Co. in China.

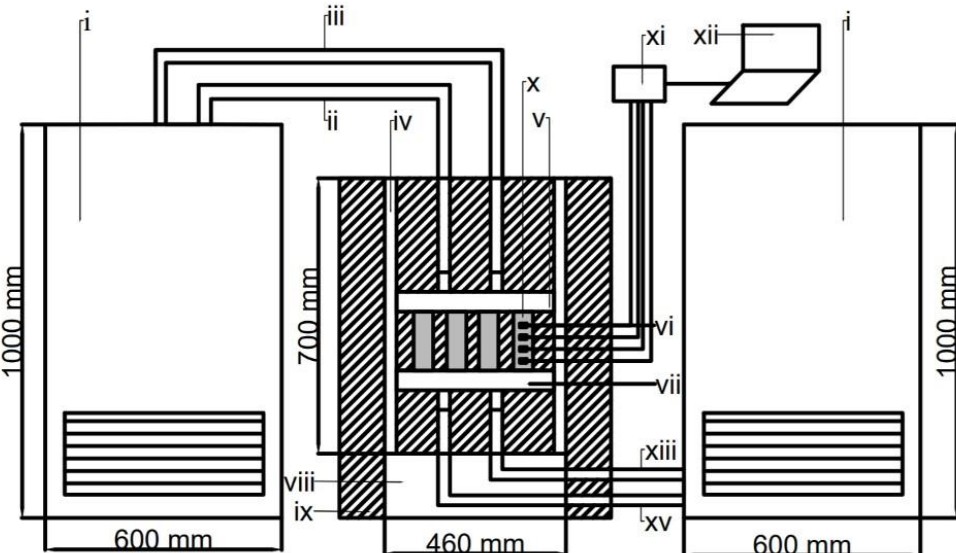

**Figure 3.** Unidirectional freezing test system. (i) Cold liquid circulator. (ii, xv) Liquid inlet. (iii, xiii) Liquid outlet. (iv) Thermostat. (v) Top plate. (vi) Temperature sensors. (vii) Bottom plate. (viii) Benchtop. (ix) Insulation material. (x) Silty clay samples. (xi) Data acquisition system. (xii) Computer.

Before the start of the tests, the samples were placed in a thermostat. In order to ensure that the samples were frozen only in the vertical direction, the samples and the temperature control plate were fully insulated. The temperature control mechanisms were at the top and bottom of the samples. During the tests, a constant temperature was maintained for the top and bottom plates. The temperature of the bottom plate was kept constant at 1 °C, while the temperature of the top plate was adjusted based on the test number.

When the tests were conducted, the temperature of the top plate was adjusted to freeze the samples in the insulation material in an axial direction, and each group of samples was frozen for 40 h. After the tests, the sample were removed and cut every 1 cm along the height direction. The total water content of each sample, which is the sum of unfrozen water content and ice content, was determined with the drying method. Next, the total water content of the layer of soil samples was determined by taking the average water content of the different samples in the group.

Figure 4 shows the frozen sample and its thermographic image. The thermal image in Figure 4 was taken with a FLIR T420 thermal imager manufactured by FLIR Systems in the US. The sample in Figure 4a was removed from the barrel immediately following the test using a pressed sample device. Although the sample was removed promptly, it was still affected by the room temperature and warmed up slightly.

### 4.4. Modeling Process

The initial stacking model for this test was established with the help of MatDEM 3.0 software. Since the subsequent forces were not considered in the numerical simulation tests and there were no special requirements for the mechanical properties of the materials, the mechanical parameters of the soil materials provided in MatDEM 3.0 were used directly for modeling. The flow of modeling is shown in Sections 4.4.1–4.4.3.

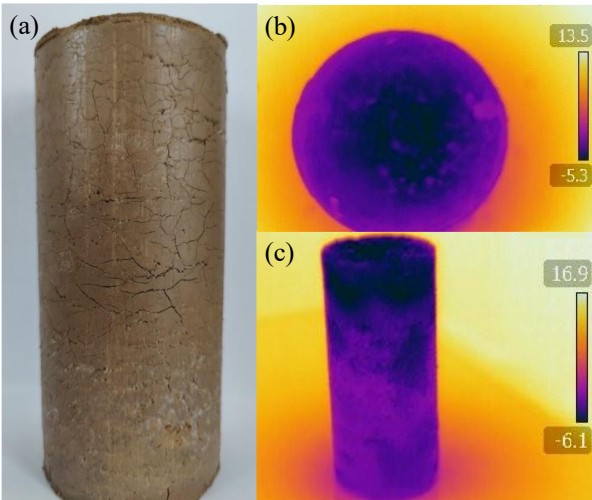

**Figure 4.** Optical and thermographic photographs. (**a**) Optical photograph. (**b**) Thermographic photograph of the top surface. (**c**) Thermal imaging photo of the front side.

### 4.4.1. Build Initial Stacking Model

To simulate the test conditions as accurately as possible and ensure computational efficiency, a soil particle element radius of 0.0015 m was used in this simulation. The particle size distribution of the soil was taken into account and an inhomogeneity coefficient of 0.3 was applied to the particle element radius. To simulate the compaction process of the soil sample in the indoor tests, the soil particles were deposited five times with a gravitational force of four times to ensure proper compactness and tightness between the particles. In this simulation, a three-dimensional initial stacking model measuring 0.075 m × 0.075 m × 0.25 m with a total of 74,086 particles was established. After compaction, the height of the model decreased from the initial 0.25 m to 0.18 m.

### 4.4.2. Cut Model

The model was cut into cylinders with the same size as the indoor test, resulting in a total of 34,190 particles in the cut model.

### 4.4.3. Set Boundary Conditions

To ensure that the simulation had the same boundary conditions as the indoor test, the front, back, left, and right boundaries were set as fixed elements with no displacement of water or heat insulation. The indoor test included fixed-temperature top and bottom plates, and to replicate these boundary conditions in the discrete element model, particles within the 0 m to 0.1 m and 0.16 m to 0.18 m height ranges were set as bottom and top plate elements, respectively. These particles, located within the top and bottom plates, were set as adiabatic and fixed elements in the model. The discrete element model simulates the indoor test by designating the top plate elements as the cold side with a temperature of −10 °C, −7 °C, and −5 °C, and the bottom elements as the warm side with a temperature of 1 °C.

### 4.5. Hydrological–Thermal Coupling Process

The simulation assigned different initial temperature values (e.g., −7 °C) to the top plate elements of the model as the boundary condition for the axial freezing process. The bottom plate elements of the model were assigned a constant temperature value of 1 °C.

To illustrate the Hydrological–thermal coupling process in the simulation, a diagram (Figure 5) is provided showing the flow of calculations within a single cycle of time steps.

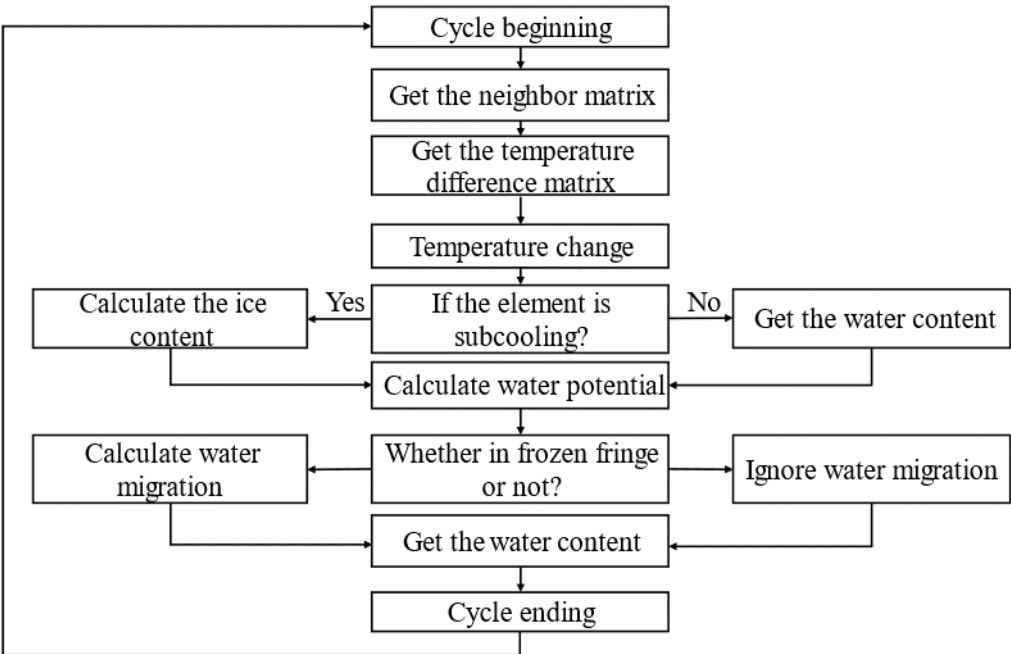

**Figure 5.** Numerical simulation flowchart.

The finite difference method was employed in the simulation to compute the temperature difference between the center element and its neighboring elements. This method enabled us to calculate the change in temperature of a particle over time based on the temperature gradients between the adjacent particles. Equation (12) was used to estimate the temperature increase of each element caused by the heat flow, providing a quantitative measure of the thermal behavior of the soil particles.

The temperature after a time step can be calculated based on the heat change of soil elements. The states of soil elements at certain times were determined by comparing their temperature with the freezing temperature. If the temperature of the soil elements was higher than the freezing temperature, it was considered to be in a non-frozen state; otherwise, it was determined to be in a frozen state.

The freezing of water in soil is not completed instantaneously, and the change in water content has a certain lag relative to the change of temperature. Therefore, the unfrozen water content in soil cannot be directly determined by temperature. Therefore, the actual unfrozen water content in the soil should be calculated based on the relationship between temperature and the ice crystal growth rate.

$$\theta_{ice}^{t+\Delta t} = \theta_{ice}^{t} + \Delta t \Delta \theta_{ice} \tag{22}$$

$$\theta_{l}^{t+\Delta t} = \theta_{l}^{t} - \theta_{ice}^{t+\Delta t} \tag{23}$$

After calculating the phase change rates $\dot{m}$ and $\Delta \theta_{ice}$ with Equation (20), the volume ice content and liquid water content of particle elements can be represented by Equations (22) and (23).

After completing the calculation of heat conduction, unfrozen water content, and ice content, the height range of the frozen fringe in the model was determined based on the average temperature of the elements at different heights in the model. Based on the range of the frozen fringe, it was determined which elements at different heights would undergo water migration, and finally the water migration of the element driven by water potential was calculated through Equation (21).

After completing the steps for thermal heat conduction, unfrozen water content, ice content, frozen fringe determination, and moisture migration, the hydrological–thermal

parameter changes within a time step were calculated. These changes were then used as input for the next time step, and the simulation continued until the desired time period was reached. By iteratively calculating these parameter changes within each time step, the simulation can model the hydrological–thermal behavior of the soil over time and provide insight into the soil's heat and water transfer mechanisms.

The parameters required in the model are shown in Table 3.

**Table 3.** Parameters required for numerical simulation.

| Parameter | Value | Unit | Parameter | Value | Unit |
|:---:|:---:|:---:|:---:|:---:|:---:|
| $a$ | 32.957 | \ | $a_w$ | 0.9985 | \ |
| $m$ | 1.825 | \ | $v_2$ | 0.3 | \ |
| $n$ | 0.236 | \ | $n_{wi}$ | 1 | \ |
| $\theta_s$ | 0.512 | \ | $\alpha$ | 0.235 | \ |
| $K_s$ | $10^{-9}$ | m/s | $\beta$ | −0.127 | \ |
| $K$ | 1.32 | W/(m·K) | $L_f$ | 334 | kJ/kg |
| $k_2$ | 0.025 | W/(m·K) | $K(T^*)_f$ | 0.0051 [40] | $s^{-1}$ |
| $C_s$ | 1800 | J/(kg·°C) | $C_w$ | 4184 | J/kg·°C |
| $C_{ice}$ | 2100 | J/(kg·°C) | $\Delta t$ | 2 | s |

## 5. Results and Discussion

### 5.1. Temperature Change and Distribution

Figure 6 shows the numerical simulation results of sample temperatures at different cold end temperatures, which show the temperature distribution of the sample at different cold end temperatures intuitively. Figure 7 shows the temperature changes during a unidirectional freezing test and numerical simulation with a cold end temperature of −7 °C. The soil's initial temperature was 10 °C. The blue frame line in Figure 6 is the boundary of the model frame in the software and has no effect on the results.

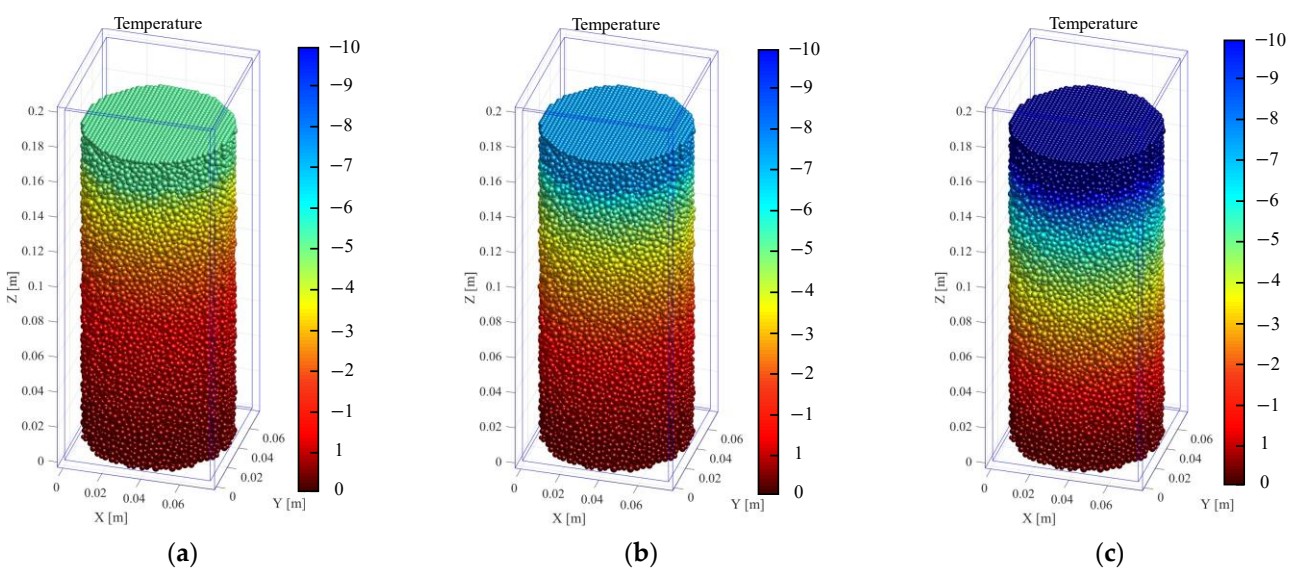

**Figure 6.** Temperature distribution of samples after freezing at different cold end temperatures: (**a**) the cold end temperature is −5 °C; (**b**) the cold end temperature is −7 °C; (**c**) the cold end temperature is −10 °C.

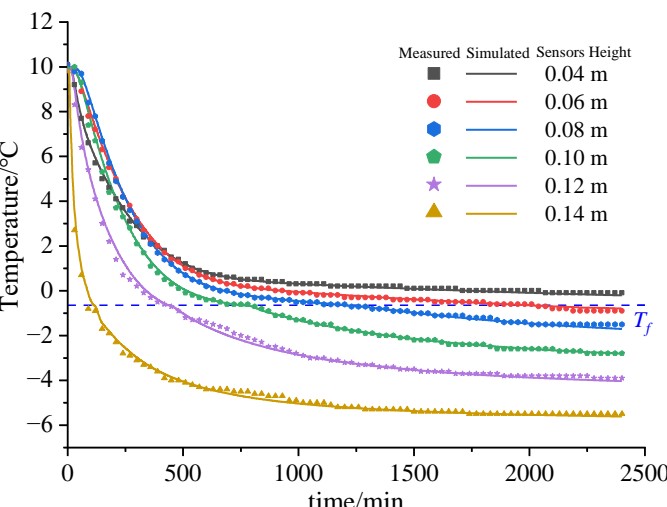

**Figure 7.** Temperature profiles at different heights of the sample during unidirectional freezing.

According to Figure 7, when the cold end temperature was −7 °C, at each point, the temperature of the soil at a height of 0.14 m experienced the most significant decrease and froze first. The sample underwent freezing from top to bottom, and the freezing process terminated when the height of the freezing front reached 0.06 m. The temporal evolution of soil temperature can be classified into three distinct phases: an initial period of rapid cooling, followed by a subsequent period of slow cooling, and, finally, a stable period. The rapid cooling period lasted approximately 400 min. During this period, the cooling rate of all parts of the sample was significant due to the large temperature gradient. The magnitude of the temperature gradient is directly proportional to the temperature of the cold end, resulting in a greater temperature change per unit time for the soil nearer to the cold end. The slow cooling period lasted roughly 400 min to 1800 min, during which the temperature gradient narrowed because of the gradual transfer of cold to the lower soil layer, and the cooling rate of the soil was slow. During the stable period, which occurred from 1800 to 2400 min at the end of the freezing process, the temperature of each soil layer remained stable with only minimal changes.

It is worth noting that the slope and trend of the curve changed after the curve crossed y = −0.6 (the freezing temperature of the soil) in the figure. Taking the temperature change curve at 0.1 m of the sample as an example, the freezing rate of the soil changed from fast to slow until 400 min, which is because of the change in temperature gradient and the exothermic influence of the phase change of the upper layer of soil in this layer. At around 400 min, the soil reached freezing temperature and the water in the soil began to solidify, during which a short stable period occurred in the temperature profile. During the short stable period, the temperature of the soil fluctuated around the freezing temperature as the liquid water released latent heat as it solidified into ice. After the phase change process of soil was basically completed, the cooling rate of the soil increased slightly. This is because the thermal conductivity and specific heat of the soil changed after the phase change, which led to the change of the thermal conductivity of that part of the soil, affecting the cooling rate of the soil. Although the thermal conductivity of the soil may vary, the temperature gradient remains the main factor affecting the cooling rate of the soil. As the temperature gradient decreased, the soil gradually entered the temperature stabilization period.

In Figure 7, the measured data and the simulated data both exhibit the same trend, accurately describing the temperature variation of the soil sample. As a result, it is feasible to incorporate the discrete element Hydrological–thermal coupling model into MatDEM 3.0 software to simulate the temperature change of soil, and the obtained results are accurate enough to reflect the temperature change of each layer of the soil sample in the unidirectional freezing more realistically.

Figure 8 shows how the position of freezing front varies with time. This also suggests that the rate of movement of the freezing front is faster for lower temperatures at the cold end. $h_{-5}$, $h_{-7}$, and $h_{-10}$ represent the position of freezing front at different cold end temperatures.

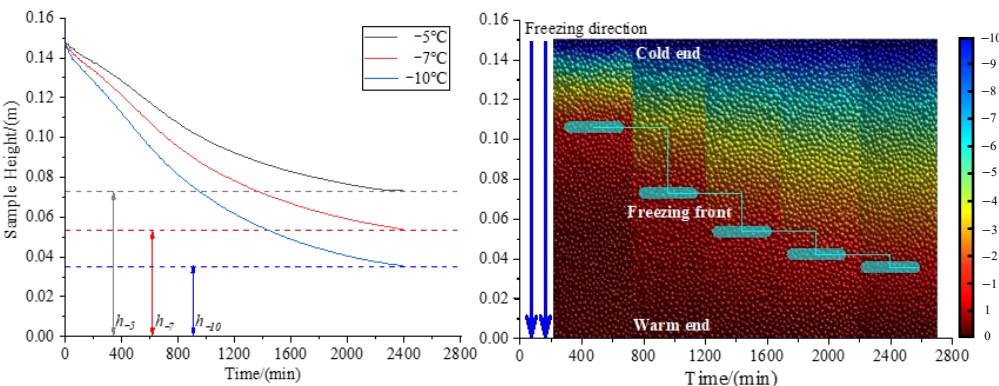

**Figure 8.** The position of the freezing front varies with time.

### 5.2. Water Migration

As can be seen from Figure 9, temperature gradient affects the location of the maximum water content and the maximum water content at the end of the test. When combined with the temperature profile during unidirectional freezing (Figure 7), the cooling rate of the elements at different heights during unidirectional freezing varies; the closer the element is to the cold end, the faster the cooling rate. Combined with the temperature curve of unidirectional freezing (Figure 7) for analysis, during the process of unidirectional freezing, the cooling rate varies among different height units, with those closer to the cold end experiencing greater cooling rates. As a result, the freezing front moves more rapidly, and the thermal state of the ice–water interface becomes more unstable, making it difficult to form a stable ice lens. Furthermore, the migration of unfrozen water from the unfrozen zone to the frozen zone is shorter, and the amount of migration is smaller.

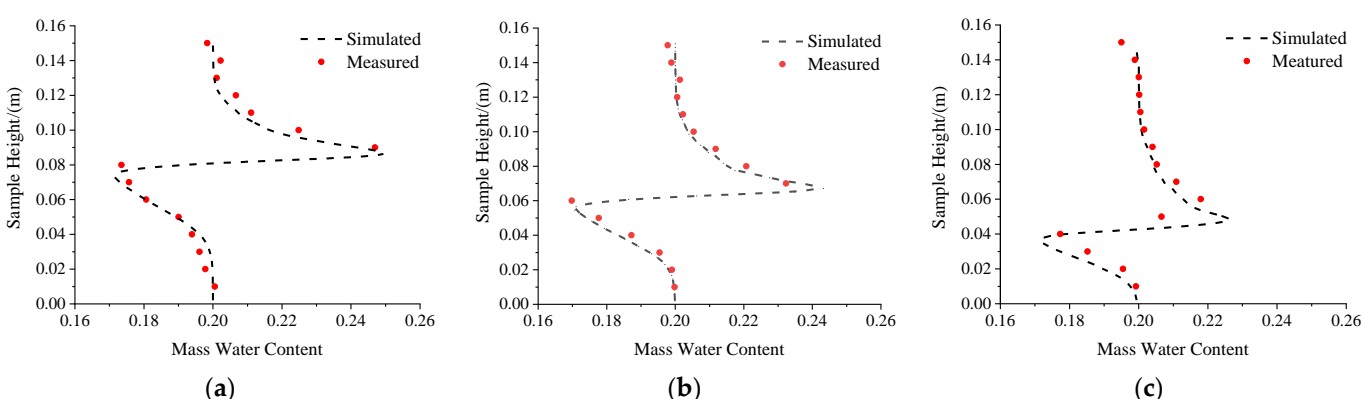

**Figure 9.** Distribution of water content along the height of the sample at different cold end temperatures: (**a**) cold end temperature of −5 °C; (**b**) cold end temperature of −7 °C; (**c**) cold end temperature of −10 °C.

For further analysis of the results, the frozen zone of soil sample can be divided into three zones based on the total water content: the rapidly freezing zone, the migrating zone, and the water-accumulating zone. In the rapidly freezing zone, the temperature of soil layer is quite low and the cooling rate is fast. Most of water in soil only freezes in situ, and almost no water migration occurs. In the migrating zone, the temperature gradient of soil is relatively thin, and the cooling rate is slightly slower. During the process of soil

cooling, the ice–water interface has a certain degree of thermal stability, and the moving speed of the freezing front slows down. These factors provide conditions for the migration of water from the unfrozen zone to the frozen zone. Under these conditions, the water content in the migration area is slightly higher compared with initial water content. In the water-accumulating zone, the freezing front stabilizes in this area with the decrease of the temperature gradient, and the water in the unfrozen area steadily migrates to the freezing front under the action of the water potential. This area has the highest water content.

Based on the assumptions made above for the frozen fringe boundary, it is clear that the larger the range contained in the segregation temperature ($T_s$) and freezing temperature ($T_f$) in the sample, the smaller the temperature gradient in the soil around the frozen fringe, and the better the conditions for water migration, the more intense the water migration within the frozen fringe. Figure 10 shows the temperature distribution of the samples along their height at different cold end temperatures. $H_{-5}$, $H_{-7}$, and $H_{-10}$ were the cold end temperatures of the samples at $-5\,°C$, $-7\,°C$, and $-10\,°C$ at 2400 min. The range (thickness of frozen fringe) is included in $T_s$ and $T_f$. The higher the cold end temperature, the greater the thickness of the frozen fringe, the more intense the water migration, and the greater the amount of water migration.

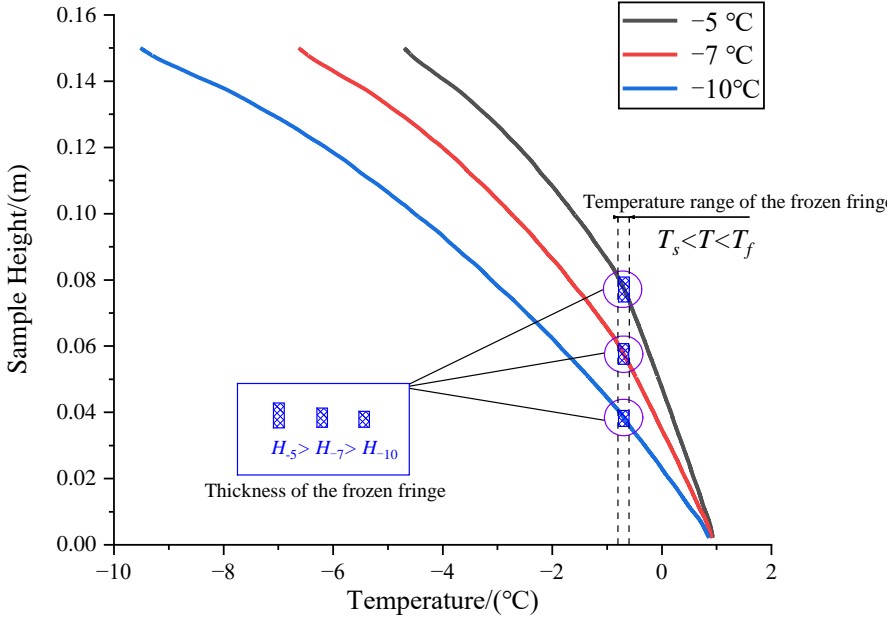

**Figure 10.** Temperature distribution along height at different cold end temperatures at 2400 min.

The red dots in Figure 6a represent the total water content of the samples. Since the spacing of the cut samples was 1 cm, the location of the maximum water content and the maximum water content obtained from the tests were not exact. The water content distribution of the samples calculated by the discrete element model is given in Figure 6, and the locations of the maximum water content and the maximum water content of the samples at different cold end temperatures are shown in Table 4.

**Table 4.** The maximum water content and its corresponding location in the samples.

| Cold end Temperature (°C) | Maximum Water Content (%) | Locations of the Maximum Water Content (m) |
|:---:|:---:|:---:|
| −5 | 25.2 | 0.073 |
| −7 | 24.3 | 0.054 |
| −10 | 22.7 | 0.036 |

## 6. Conclusions

Through the analysis of the discrete element numerical simulation results and their comparison with the indoor test results, the following conclusions were obtained:

1.  The discrete element hydrological–thermal coupling model established by introducing thermal conduction, water migration equation, and the relationship between temperature and unfrozen water content is logically tight and has no harsh assumptions. The model can be further developed and customized to meet specific requirements and applications;

2.  Through indoor tests and discrete element numerical simulation, this paper found that when silty clay undergoes unidirectional freezing, the unfrozen water in the soil will migrate to the freezing front, and the larger the temperature gradient, the smaller the amount of migration. This article can correctly describe the thermal conduction and moisture migration process of silty clay under unidirectional freezing conditions;

3.  The simulation results obtained by the discrete element hydrological–thermal coupling model established in this paper can describe the hydrological–thermal parameters of soil samples. The parameters such as water content and temperature obtained by numerical simulation can be accurate to each element. After the stress field coupling is added to the subsequent research, the factors such as cracks and consolidation caused by frost heave can be taken into account.

The discrete element hydrological–thermal coupling model for unsaturated soil developed in this paper can realistically reflect the changes of hydrological–thermal parameters when freezing unsaturated soil in a unidirectional closed system, and can construct boundary conditions can be adjusted to simulate different conditions. This paper provides a new idea for the application of a multi-field coupled model of the discrete element method in cold region engineering.

**Author Contributions:** Conceptualization, W.S.; arrangement of test data, S.Q.; formal analysis, W.S., S.Q.; funding acquisition, W.S.; writing—original draft, W.S., S.Q., writing—review and editing, W.S., S.Q., Y.G. All authors have read and agreed to the published version of the manuscript.

**Funding:** We thank the National Natural Science Foundation of China (Grant No. 41641024), the Carbon Neutrality Fund of Northeast Forestry University (CNF-NEFU) and the Science and Technology Project of Heilongjiang Communications Investment Group (Grant No. JT-100000-ZC-FW2021-0182) for providing financial support and the Field Scientific Observation and Research Station of the Ministry of Education—Geological Environment System of Permafrost Areas in Northeast China (MEORS-PGSNEC).

**Data Availability Statement:** All data generated or analyzed during this study are included in this published article.

**Acknowledgments:** We would like to thank all staff members who contributed to this study who are not named here.

**Conflicts of Interest:** The authors declare no conflict of interest.

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
