# Peer review of "Hydrological–Thermal Coupling Simulation of Silty Clay during Unidirectional Freezing Based on the Discrete Element Method"

_water, doi:10.3390/w15071338_

Round 1

Reviewer 1 Report (Previous Reviewer 3)

The authors answered all my questions.

Author Response

Reviewer 2 Report (Previous Reviewer 1)

From the content of the newly submitted manuscript, the author has made a lot of modifications through reading the literature, and the accuracy of the data and the content shown in the pictures has been greatly improved, which is in good agreement with the previous research results. The results of the discrete element analysis and the results of the indoor one-way test also have good consistency. At the same time, other aspects have also made corresponding modifications, so that the quality of the manuscript has been improved a lot, meeting the publishing requirements, it is recommended to consider accepting!

Author Response

This manuscript is a resubmission of an earlier submission. The following is a list of the peer review reports and author responses from that submission.

Round 1

Reviewer 1 Report

Manuscript Number: water-2065917-peer-review-v1 Comments

Title: Discrete element 3D simulation of heat-moisture coupling for unsaturated silty clay axial freezing test

According to the title and the content described in the paper, the paper mainly conducts experiments on the migration law of unfrozen water in unsaturated silty clay under freezing conditions and three-dimensional discrete element numerical simulation analysis. The mathematical model is mainly introduced into the discrete element numerical simulation software for simulation calculation, and compared with the indoor axial freezing test results. Finally, some results about the moisture transfer process are obtained. This is a very meaningful topic, but there are key problems in the modeling process and simulation results in the manuscript.

Specific review comments are as follows:

(1) The theoretical models are all the description of unsaturated soil parameters, and there is no analysis and derivation process of moisture heat coupling mechanism in soil freezing process.

(2) As far as the test in this paper is concerned, the results of 3D and 2D are of little significance, because the soil sample is frozen in a uniform layer from top to bottom. Secondly, from the perspective of manuscript reasoning and test results, there is no difference between 3D, 2D and one-dimensional calculation results, so, the manuscript title is certainly inappropriate, it means a little high on purpose.

(3) There is a problem with the drawing, which is not standard. For example, the ordinate in Figure 6 should be consistent with the freezing direction of the sample for comparative analysis.

(4) The calculation result is obviously wrong, which may be caused by errors in the calculation model and assumptions. For example, Figure 7 (b) shows that the water content of the upper top increases with the decrease of temperature. The lower the temperature, the greater the water content of the upper top, which is obviously wrong. In fact, the maximum water content is generally near the frozen layer.

(5) The initial water content shown in Figure 7 is 0.2, and the maximum water content and the location change of the frozen layer during the freezing process should be marked.

(6) As for the experimental research on the freezing process of cylindrical soil, the reviewers have consulted many documents and have also obtained good calculation results. It is recommended that the author carefully read and analyze the research results of relevant documents to enhance the reliability and innovation of the manuscript research.

Reviewer 2 Report

Introduction is recommended to be partly improved, because the topic of modelling of moisture migration movement at freezing was internationally, widely and deeply studied, and it is reasonable to mention more references.

The main thing in the paper, coupling thermal-migration model, remains partly unclear. The thermal conductivity equation is proposed, but how the heat of phase transfer "water-ice" is taken into calculations remains unclear.

A migration potential is mentioned, but relation to the well-known segregation potential (Konrad, 1981) is not shown by authors. 

???? is the internal pressure of the pore in equation (13), and it is not clear how authors differ it from ?? which is the air pressure in the pore.

It is suggested to use ?s of equation (14), but it was shown before (equations 5-10) that diffusivity of moisture is actually more complicated. If Ks is used after, what purpose to introduce equations 5-10?

Reviewer 3 Report

In this paper, based on the discrete element method, a heat-moisture coupling model for powdered clay in the case of unidirectional freezing is proposed. Through indoor tests and numerical simulations, the changes of soil physical parameters in silty clay under axial freezing conditions are investigated and the proposed numerical model is validated. Indoor tests and numerical simulations show that when axial freezing occurs in chalky clay soils, unfrozen water in the soil migrates from unfrozen to frozen areas, and the amount of water migration increases with the decrease in freezing temperature. The results of indoor tests and numerical simulations fit well and prove the usability of the model. The logic of the whole paper is rather good, and the way of establishing the heat-moisture coupling model in this paper is relatively novel. The use of the discrete element method is a good fit for fractured permafrost soils, which deserves recognition. In addition, this paper is based on the study of changes in basic physical properties of soil during freezing, a direction that is in line with the trend of research related to permafrost engineering, which is much appreciated. The reviewer only has some minor recommendations as follows:

1. In the abstract, you can include possible applications of the methods used in the article. 

2. On page 3, the fractional equation in equation (1) is written differently than the other equations in the article. Please proofread the entire article carefully to eliminate such errors.

3. In the title of Chapter 2, page 3, "Theorys and Methods" should be "Theories and Methods".

4. In equation (3) on page 4, what do  and  represent respectively?

5. On page 8, subsection 3.4.1, a number of formatting errors were found.

6. In Figure 2 on page 9, please check the consistent definition of “moisture content” in the article and in the graphs, where both "water content" and "moisture content" are stated. Please also check for similar problems throughout the article.

7. A discrete element software called MatDEM was used in the article. How did this software help your research and is there any literature to support the applicability of this software?